# Hierarchical Multi-Objective Optimization for Dedicated Bus Punctuality and Supply–Demand Balance Control

**DOI:** 10.3390/s23094552

**Published:** 2023-05-07

**Authors:** Chunlin Shang, Fenghua Zhu, Yancai Xu, Xiaoming Liu, Tianhua Jiang

**Affiliations:** 1College of Transportation, Ludong University, Yantai 264025, China; itsshang@ldu.edu.cn (C.S.);; 2State Key Laboratory of Multimodal Artificial Intelligence Systems, Institute of Automation, Chinese Academy of Sciences, Beijing 100190, China; 3Beijing Key Laboratory of Urban Road Traffic Intelligent Control Technology, North China University of Technology, Beijing 100144, China

**Keywords:** intelligent transportation system, hierarchical multi-objective optimization, driving speed decision-making, dedicated bus, Lagrangian multiplier method

## Abstract

Public transportation is a crucial component of urban transportation systems, and improving passenger sharing rates can help alleviate traffic congestion. To enhance the punctuality and supply–demand balance of dedicated buses, we propose a hierarchical multi-objective optimization model to optimize bus guidance speeds and bus operation schedules. Firstly, we present an intelligent decision-making method for bus driving speed based on the mathematical description of bus operation states and the application of the Lagrange multiplier method, which improves the overall punctuality rate of the bus line. Secondly, we propose an optimization method for bus operation schedules that respond to passenger needs by optimizing departure time intervals and station schedules for supply–demand balance. The experiments were conducted in Future Science City, Beijing, China. The results show that the bus line’s punctuality rate has increased to 90.53%, while the retention rate for platform passengers and the intersection stop rate have decreased by 36.22% and 60.93%, respectively. These findings verify the effectiveness and practicality of the proposed hierarchical multi-objective optimization model.

## 1. Introduction

Due to the rapid increase in traffic demands, traffic congestion has become a major problem worldwide. To combat this issue, various methods have been proposed, with bus priority being considered the most effective method. This has led to transit-oriented development (TOD) [1] being adopted as a fundamental strategy by many countries, resulting in significant progress over decades of development. Millions of kilometers of dedicated bus lanes have been constructed, and the urban bus dedicated lane network is taking shape. These achievements have reduced the impact of non-special vehicles on public transport and have significantly improved the service level of dedicated buses.

The problems related to dedicated bus lanes are also widely studied, attracting thousands of scholars worldwide. Some innovative ideas have been proposed to optimize and improve the service level of dedicated bus lanes, such as reducing per capita delays, decreasing parking rates, and increasing bus operation speeds. However, focusing solely on punctuality as a single target cannot adequately reflect the reliability and stability of buses. Combining this understanding with the literature [2,3,4], we find that multi-objective optimization can overcome the limitations of single-objective optimization. However, the multi-objective weighted processing method often compromises punctuality performance due to the consideration of other objectives. Moreover, there is a lack of studies that consider the overall punctuality service level and the supply and demand service level of the entire bus line. To address this gap, this paper proposes a hierarchical multi-objective optimization model. The first layer focuses on optimizing the bus guidance speed to meet punctuality requirements. The objective is to achieve the highest comprehensive punctuality rate for the entire bus line. Considering the constraints posed by multiple stations and intersections, the Lagrange multiplier method was employed to determine the optimal line guidance speed. The second layer deals with the optimization of bus schedule scheduling to achieve a supply and demand balance. This layer establishes a supply and demand balance model by continuously monitoring the number of passengers waiting at the platform and the number of passengers on the bus. Finally, we obtained the optimal departure time interval using the genetic algorithm. This article’s algorithm optimizes bus operations at multiple levels, including punctuality rate and passenger service level, by monitoring road traffic and bus operation status and utilizing data-edge computing methods. It is important to note that reliable infrastructure investment is a prerequisite for implementing this algorithm. The optimized approach increases the proportion of public transportation trips, enhances the efficiency and reliability of trunk bus transportation, and ensures the successful implementation and application of public transportation priority policies. In summary, the main contributions of this paper include:

(1) Existing research on dedicated buses often separates the optimization of the punctuality service level and the supply–demand balance. However, there is a lack of correlation and collaboration between these two goals. To address this issue, a hierarchical multi-objective optimization model has been proposed to optimize the punctuality service level and achieve a balance between supply and demand for dedicated buses.

(2) Many existing studies employ weighted processing to tackle the multi-objective optimization problem. However, this approach can potentially result in the main objective being influenced by other objectives. To overcome this limitation, a hierarchical multi-objective speed decision-making method utilizing the Lagrangian multiplier method is proposed. This method prioritizes ensuring the comprehensive punctuality rate of the entire bus line and subsequently optimizes stop times at intersections while achieving a balanced distribution of guidance speeds.

(3) The traditional method of optimizing bus operating schedules, which solely relies on travel time, fails to consider the crucial aspect of balancing passenger demand with bus capacity. To address this issue, a bus operation schedule optimization method has been proposed to achieve a balanced relationship between bus capacity and passengers waiting at stations. This approach helps overcome inconsistencies between bus schedule settings and passenger travel demand.

The structure of this paper is as follows: Section 2 introduces the current transit priority control strategy. Section 3 outlines the assumptions and research framework used in the article. Section 4 details the research method of bus speed optimization, including mathematical descriptions of the bus operation, the multi-objective speed-decision-making model, and the Lagrange multiplier method for speed optimization. Section 5 explains the research method of bus schedule optimization, including bus departure intervals and station schedule optimization. Section 6 presents the testing results. Finally, Section 7 provides the basic conclusions and discusses the possibility of future research.

## 2. Literature Review

This part introduces the current transit priority control strategy, which mainly covers three parts: transit signal priority, speed guidance optimization, and schedule optimization. Then we summarize the current research status. Figure 1 provides an overview of the research interests and methodologies of select scholars.

### 2.1. Transit Signal Priority

In order to reduce the waiting time of buses at signal-controlled intersections and enhance bus operating efficiency, several scholars have conducted research on transit priority control, starting with intersection signal priority. Tarikul developed a dynamic priority control system that balances signal control demands and transit priority, aiming to reduce the average person delay at intersections [5]. Qiao proposed an optimization model for transit signal priority at a signalized intersection based on the phase clearance reliability index, with the objective of minimizing average person delay [6]. Truong presented an advanced transit signal priority (ATSP) control model that considers the arrival distributions of buses at downstream intersections when providing priority at upstream intersections, resulting in an approximate 10% improvement in bus line efficiency [7]. Li reduced the average person delay and improved the traffic efficiency of the trunk line by implementing a signal priority control method based on coordinating green waves [8].

Scholars have also studied transit priority from the perspective of pre-signal settings, considering that it may reduce the operating efficiency of social vehicles. He aimed to reduce the average person’s delay by prioritizing bus signals through adaptive control and pre-signal settings [9]. Liang analyzed the queuing situations at bus intersections using the distributed wave theory and designed a pre-signal control algorithm for transit priority based on queuing length to achieve non-stop traffic at intersections [10]. Bie developed a coordinated control algorithm between the main signal and pre-signal of bus intersections based on the pre-signal of transit priority, thereby reducing the impact of transit priority phases on social vehicle efficiency [23]. While these studies reduced bus delays at intersections to some extent, they failed to improve the reliability of bus punctuality, making it difficult to further enhance the punctuality service level of buses.

### 2.2. Bus Speed Guidance

With the advancement of vehicle–road cooperative technology, real-time speed guidance for bus vehicles has become possible. This development has prompted scholars to explore improvements in bus service levels through the lens of speed guidance. Khaled achieved further reductions in bus delays by implementing a strategy of early braking at red lights and extending green lights based on speed guidance while incorporating transit signal priority [24]. Shu solved direct bus and left turn bus priority control using speed guidance under the premise of transit signal priority, improving bus traffic efficiency at intersections [11]. Chiara determined different speed guidance strategies by analyzing the priority of bus formation and independent buses, reducing average person delay [12]. Deng proposed a dynamic real-time speed guidance model to mitigate operation delays of bus lines caused by signalized intersections and uneven road conditions [13].

Scholars also studied speed guidance from the perspective of improving bus punctuality since it enhances the bus operation reliability. Takashi developed an inter-station control strategy that coordinates speed guidance and signal control based on punctuality demands for bus arrivals, leading to improved reliability of bus arrivals [14]. Yan proposed a real-time bus speed control strategy to minimize the mean absolute error of bus headway due to unstable bus arrival times, significantly enhancing the punctuality reliability of bus arrivals [15]. Zhang analyzed bus trajectory data and implemented guidelines to achieve a balanced distribution of bus headway, thereby improving bus operation reliability [16]. While these studies have made improvements in enhancing the reliability of bus operations, optimizing speed guidance solely based on a single road section may lead to local optimization but poor overall conditions.

### 2.3. Bus Schedule Optimization

The key to improving bus punctuality and operational efficiency involves setting the bus schedule, prompting scholars to focus on its optimization. Li developed a public transport scheduling model for a microsystem that aims to minimize passenger waiting time while maximizing the number of passengers per bus. This is achieved by optimizing departure intervals and utilizing both traditional and rapid buses simultaneously [17]. Gkiotsalitis constructed an optimization model for bus travel schedules based on passenger demand and travel time expectations [18]. Shang proposed an evaluation model that takes into account both passenger satisfaction and traffic efficiency. This model was used to optimize and adjust the bus operating schedule, and its effectiveness was demonstrated through a case study conducted in Beijing [25]. Banerjee used a school bus as an example and proposed a scheduling model aimed at optimizing the operational efficiency of bus schedules [19]. Teng established a multi-objective optimization model to balance departure headway, reduce the number of vehicles used, and lower electric bills. This model optimized bus operation schedules under multiple constraints, including limits on the range of departure intervals, the number of available vehicles, and bus endurance mileage at different periods [20]. To optimize the bus schedule, Liu constructed a super-efficient DEA model based on indicators of passenger waiting time and congestion [21]. Ma proposed a dynamic schedule optimization scheme based on the correlation of passenger time demand and travel time between stations, utilizing bus GPS data and IC card data [26]. Zhang developed an optimal design model to minimize passenger transfer waiting time, considering the importance of transfer stations and different travel time utility values for passengers with different travel purposes, optimizing the bus schedule using genetic algorithms [22]. While these studies improved bus operation efficiency and passenger satisfaction to some extent, they rarely optimized and adjusted bus capacity allocation and schedule schemes from the perspective of passenger travel demand, making it challenging to optimize passenger platform retention.

### 2.4. Summary

Existing research has made progress in optimizing bus service levels at various levels, yielding some favorable outcomes. However, most punctuality controls focus on optimizing a single section, which overlooks the optimization of bus punctuality across the entire line. As a result, there is a risk of local optimization with overall poor performance, affecting bus punctuality service levels. Furthermore, existing research often neglects the correlation between passenger travel demand and bus capacity allocation, which can lead to a mismatch between supply and demand, thereby affecting the quality of bus service.

## 3. Outline of Research Framework

### 3.1. Assumptions

(1) All signal-controlled intersections along the route lack a bus signal priority strategy, and bus vehicles are not granted signal priority at intersections;

(2) All bus stations on the route are designed as single-line bus stations, and there are no instances where multiple lines share a single bus station;

(3) There is a signal-controlled intersection between every two bus stations on the line;

(4) All passengers on the bus are required to swipe their cards at the front door when boarding and at the rear door when disembarking.

Assumption (1) aims to enhance the algorithm’s general applicability in this paper by considering that the majority of dedicated bus lanes do not implement a bus signal priority strategy. Assumption (2) primarily seeks to minimize the impact of multiple bus lines overlapping in the analysis of this method. Assumption (3) aims to establish a standardized bus operation environment and minimize the influence of variations in the bus operation environment on the method proposed in this paper. Assumption (4) aims to standardize the information collection method for passengers boarding and alighting from the bus, ensuring consistent data quality.

### 3.2. Research Framework

In order to improve the punctuality service level and supply–demand service level of dedicated buses, we analyzed the signal data, bus data, and passenger flow data, leading to the construction of a hierarchical multi-objective optimization model. From the perspective of bus operating reliability and comfort, the first layer of the multi-objective optimization model for intelligent decision-making of bus speed was developed. This layer primarily encompasses a mathematical description of bus operation, a multi-objective speed-decision-making model, and the Lagrange multiplier method for speed optimization. By considering multiple target constraints, such as the reliability of bus arrivals, non-stop operations at intersections, and smooth speed distribution, the punctuality service level of public transport has been improved. According to passenger travel needs and bus capacity allocation, the second layer of the multi-objective optimization model for the bus supply and demand balance was constructed. This layer mainly includes bus departure interval optimization and station schedule optimization. From the balanced analysis of the passenger stasis of the platform and the passenger capacity of the bus, in this paper, we realize the optimization of the bus departure time interval. Moreover, we optimize the bus station schedule by combining it with an analysis of the driving characteristics between the bus stations.

The collection of traffic data is not the primary focus of this paper. Therefore, existing methods were employed to gather multi-source traffic data, including intersection signal control data, bus dwell, and travel data, as well as passenger station waiting, boarding, and alighting data.

Figure 2 presents the research framework of this paper. As shown in Figure 2, the hierarchical multi-objective optimization mainly includes a speed optimization layer and a schedule optimization layer. The speed optimization layer selects the best bus guidance speed in multi-objective and multi-constrained states; the schedule optimization layer is mainly conducted through schedule adjustments to achieve a balance between passenger travel demand and bus operation configuration.

## 4. Optimization of Bus Speed

Figure 3 shows the schematic diagram of bus speed guidance. We can see that when considering only the constraints of the bus passing through intersection 1 (without stopping and arriving at platform 2 on time), both scheme 1 and scheme 2 can fulfill the requirements. However, scheme 2 will pause at intersection 2 due to the influence of the dwell time at platform 2, resulting in the bus being unable to arrive at platform 3 on time. Scheme 1 can satisfy the requirements of arriving at both platform 2 and platform 3 on time. It shows that the improvement in the bus punctuality service level should consider not only the scene constraints between bus stations but also the speed guidance constraints from the perspective of route optimization. To this end, this paper studies the optimal bus speed guidance strategy from the perspective of route optimization. Moreover, $t_{\mathrm{gs}},t_{\mathrm{ge}}$ is the green light start/end time, *T* is the bus schedule arrival time, and $V_{\max}$ is the max speed of the road section.

Considering that the majority of existing scheduled operation plans are based on minute intervals, this paper defines buses arriving within one minute of the scheduled platform time as punctual. For instance, if the scheduled arrival time is 8:05, buses arriving at the station between 8:05 and 8:06 are considered punctual.

From analyzing Figure 3, it can be seen that the buses are mainly affected by bus dwell time and signal timing. As a result, bus speed guidance can be achieved by considering the analysis of bus dwell time and the current state of intersection signals. Since there have been many research studies [27,28,29,30] conducted by scholars worldwide on the prediction of bus dwell time, we adopted the research method outlined in reference [28] to obtain the predicted value $t_{pre}^{i}$ of the bus dwell time of platform *i*.

### 4.1. Mathematical Description of Bus Operation

The main objective of this section is to obtain the corresponding state of the bus during all stages of operation, which primarily includes the stages of bus dwell, road operation, and intersection waiting.

When the bus leaves the i-th platform, the departure time can be defined as $t_{is}$; moreover, when the departure time is determined, the moment $t_{ia}$ when the vehicle arrives at the intersection is mainly affected by the speed of the off-station bus, so we can construct a set $V_{c}^{1i}$ of vehicle speeds that satisfy vehicles passing through intersections without stopping. Moreover, if $V_{c}^{1i}=\emptyset$, it means that the bus needs to wait for a red light at the intersection. At this time, the stopping–waiting time of the vehicle at the intersection is $t_{iwait}^{c}=g_{start}^{i}-t_{ia}$.

Considering the requirement for the bus to arrive punctually, when the vehicle drives away from the intersection, we need to determine a driving speed in front of the station to ensure that the vehicle arrives on time. Thus, we can construct a set $V_{c}^{2i}$ of vehicle speeds that satisfy vehicles arriving at the station on time. Moreover, if $V_{c}^{2i}=\emptyset$, it means that the bus cannot arrive on time. Therefore, the error of the arrival time is $\Delta t_{a}^{i}=\min\left(\left|T_{i}-t_{ia}\right|,\left|T_{i}+60-t_{ia}\right|\right)$.

The bus station departure time is mainly determined by the dwell time at the bus station and the arrival time of the bus, and we know that the bus arrival time is determined by the state of the bus at the upstream intersection. The dwell time mainly includes the predicted value $t_{pre}^{i}$ and the forecast error compensation value $\Delta t_{dwell}^{i}$ so we can obtain the mathematical description of the bus operation.
(1)$$\left\{\begin{array}{c} t_{ia}=t_{1a}+\sum_{j=1}^{i-1}\left[f\left(v_{c}^{1j},v_{c}^{2j}\right)+t_{lost}^{j}\right]\\ t_{lost}^{i}=t_{pre}^{i}+\Delta t_{dwell}^{i}+t_{iwait}^{c}\\ f\left(v_{c}^{1j},v_{c}^{2j}\right)=\frac{s_{j}^{1}}{v_{c}^{1j}}+\frac{s_{j}^{2}}{v_{c}^{2j}} \end{array}\right.\;$$

$t_{1a}$ is the arrival time of the bus at the first station, $t_{lost}^{i}$ is the dwell loss time of the bus, $s_{i}^{1},s_{i}^{2}$ is the distance between the *i*-th intersection and the upstream and downstream platform sections.

### 4.2. Multi-Objective Speed-Decision-Making Model

Optimizing a single objective does not guarantee an overall improvement in the level of public transport services. To address this, this paper presents a hierarchical multi-objective optimization model. The model includes the following objectives: achieving the highest comprehensive punctuality rate for the bus lines as the first-level objective, minimizing the intersection stopping rate as the second-level objective, and achieving the most balanced distribution of guidance speed as the third-level objective.

First, this paper uses the error of the arrival time to analyze the punctuality of the entire line; the discriminant function of punctual arrival can be determined as follows:(2)$$\left\{\begin{array}{c} M=\sum_{x=1}^{i}m_{x}\\ m_{i}=\left|\left\lceil \frac{\Delta t_{a}^{i}}{\Delta t_{\max}}\right\rceil -1\right| \end{array}\right.$$

The number of punctual bus arrivals of the entire route is M,0≤M≤N, the number of bus stations in the line is *N*, and $Deltat_{\max}$ is the historical maximum error. We use the round-up function to obtain the punctual arrival state $m_{i}$ of the bus; if $m_{i}=0$, it is non-punctual; if $m_{i}=1$, it is punctual.

The traditional bus punctuality index considers only whether the arrival time falls within the specified constraint range, without considering the magnitude of the non-punctual arrival error and its distribution state. As a result, it becomes challenging to comprehensively evaluate and optimize bus punctuality control with accuracy. For this reason, this paper selects the variance of the arrival error $D\left(\Delta t_{a}\right)$, the average value of the arrival error $\bar{\Delta t_{a}}$, and the proportion of the number of non-punctual arrivals N−MN−MNN as the index factors of the comprehensive punctuality rate and the sigmoid function is used to normalize the three as follows:(3)σ1=2sigmoidDΔta−12=1−e−DΔta1+e−DΔtaσ2=2sigmoidΔta¯−12=1−e−Δta¯1+e−Δta¯σ3=1+e−11−e−1×1−e−N−MN−MNN1+e−N−MN−MNN

$\sigma_{1},\sigma_{2},\sigma_{3}\in \left[0,1\right]$, when $D\left(\Delta t_{a}\right)=0$, $\sigma_{1}=0$; when $D\left(\Delta t_{a}\right)=\infty$, $\sigma_{1}=1$; when $\bar{\Delta t_{a}}=0$, $\sigma_{2}=0$; when $\bar{\Delta t_{a}}=\infty$, $\sigma_{2}=1$; when M=N, $\sigma_{3}=0$; when M=0, $\sigma_{3}=1$. Based on (4) and (5), it can be observed that they correlate with $V=\left(V_{c}^{1\left(i-1\right)},V_{c}^{2\left(i-1\right)}\right)$.

The optimization objective for the comprehensive punctuality rate of bus lines is defined as (4). It is evident that there is a constraint that the guidance speed must satisfy at this time:(4)$$\left\{\begin{array}{c} J_{1}=\min\left(\phi_{1}\sigma_{1}+\phi_{2}\sigma_{2}+\phi_{3}\sigma_{3}\right)=\min\left(H\left(V\right)\right)\\ \delta_{1}\left(V\right)=t_{ia}-T_{i}\\ 0\leq \delta_{1}\left(V\right)\leq 60 \end{array}\right.$$

This article analyzes the bus stop times at intersections during the bus operation. The stop times, denoted as *K*, are directly linked to the set of guidance speeds $\left(V_{c}^{1i},V_{c}^{2i}\right)$. Therefore, it is possible to establish a judgment model of the bus state in combination with the execution state of the intersection signal control.
(5)$$\left\{\begin{array}{c} K=G\left(V_{c}^{1\left(i-1\right)},V_{c}^{2\left(i-1\right)}\right)=\sum_{j=1}^{i}k_{j}\\ k_{i}=\left\lceil \frac{t_{iwait}^{c}}{C^{i}}\right\rceil \end{array}\right.$$

We use the round-up function to obtain the bus stop state $k_{i}$ at the intersection; if $k_{i}=0$, it means that there is no stop at the intersection; if $k_{i}=1$, it means a stop is required at the intersection.

Therefore, we can determine the objective of minimizing the stop times at intersections along the entire bus route, as well as the constraints that the guiding speed must meet:(6)$$\left\{\begin{array}{c} J_{2}=\min\left(K\right)=\min\left(G\left(V\right)\right)\\ \delta_{2}\left(V\right)=t_{ge}^{i}-t_{ia}-t_{pre}^{i}-\Delta t_{dwell}^{i}-\frac{s_{i}^{1}}{V_{c}^{1i}}\\ 0\leq \delta_{2}\left(V\right)\leq g_{i} \end{array}\right.$$

Each road section is associated with a specific guidance speed, and the guidance speeds vary along the entire bus route. However, if there is a significant difference in guidance speeds between road sections, it can lead to a poor driving experience or increase the risk of traffic accidents. To ensure a balanced distribution of bus speeds, it is essential to optimize and analyze the discrepancies in guidance speeds across each road section.

We can determine the guide speeds of the route based on the above analysis; we obtain the balanced distribution speed control scheme by the variance calculation.
(7)$$J_{3}=\min\left(D\left(V\right)\right)=\min\left(D\left(V_{c}^{1\left(j-1\right)},V_{c}^{2\left(j-1\right)}\right)\right)$$

### 4.3. Lagrange Multiplier Method for Speed Optimization

According to (4), (6) and (7), the solution to the optimal speed guidance is essentially a multi-objective hierarchical optimization problem with inequality constraints. The first-level objective is to achieve a comprehensive punctuality rate that ensures the bus arrives on time. The second-level objective is to minimize stop times at intersections, optimizing the overall bus operation service level while still meeting the requirement of timely arrivals at stations. The third-level objective focuses on achieving a balance in guide speeds, reducing speed discrepancies among road sections, and enhancing bus operation stability while maintaining punctual arrivals and non-stop intersections. Combined with the research in the literature [31,32,33], this paper adopts the Lagrangian multiplier method (Algorithm 1) to solve the problem in a hierarchical manner.
**Algorithm 1** Lagrangian multiplier method.**Input I: Guide speed of each section *V*****Output I: Optimization results Vfinal**
1: According to (4), determine the first-level optimization goals and constraints:J1=minHVs.t.0≤δ1V≤Tzhuni0≤V≤Vmax
2: Determine the Lagrangian multiplier:L1V,λ1,λ2=HV+λ1δ1V−Tzhuni+η12  +λ2V−Vmax+η22
λ1,λ2 are Lagrange multipliers, and the new η1,η2 are relaxation variables. The purpose is to change the inequality constraint into the equality constraint after relaxation; note that η1,η2≥0.
3: Solve the speed combination VL1:∂L1∂V=0,∂L1∂λ1=0,∂L1∂λ2=0,∂L1∂η1=0,∂L1∂η2=0
4: According to (6) and (7), the third-level optimization goal can be regarded as a constraint of the second-level optimization goal, so we can determine the second-level optimization goal and constraints:J2=minGVL1J3=minDVL1s.t.0≤δ2VL1≤gi
5: Determine the Lagrangian multiplier:L2VL1,λ3=GVL1+λ3DVL1+λ4δ2VL1−gi+η32
λ3,λ4 are Lagrange multipliers, and the new η3 is relaxation variable. The purpose is to change the inequality constraint into the equality constraint after relaxation, note that η3≥0.
6: Solve the speed combination Vfinal:∂L2∂VL1=0,∂L2∂λ3=0,∂L2∂λ4=0,∂L2∂η3=0
end


## 5. Optimization of Bus Schedule

The setting of the bus schedule involves coordinating the balance of bus service levels and bus operating costs [34,35]. Bus schedules not only affect the punctuality of bus arrivals but also influence the passenger capacity of bus lines. Thus, it is crucial to set bus schedules scientifically. However, existing bus operation schedules are often based on empirical methods and involve subjective factors. This can lead to issues such as underutilized bus capacity and empty runs caused by excessive departure frequency, as well as overcrowded buses and platforms resulting from low departure frequency. These problems lead to resource wastage and a decline in travel service levels. Therefore, this paper optimizes bus operation schedules by considering the optimization of departure time intervals and station schedules.

### 5.1. Bus Departure Intervals Optimization

The primary reason for the retention of passengers who are waiting at the platform is the mismatch between the free load of the bus and the number of passengers who are waiting. Considering that there is an upper limit of $Q_{\max}$ in the maximum passenger load of the bus, the optimization of the passenger flow mainly depends on the analysis of the number of people waiting. By setting a passenger flow detector at the bus station and using 1 min as the statistical interval, the number of passengers arriving at platform *i* is $q_{i}\left(t\right)$, and the number of passengers waiting at the platform between the two bus stations can be obtained as follows:(8)$$Q_{wait}^{i}=\int_{0}^{t_{p}}q_{i}\left(t\right)dt$$
$t_{p}^{i}$ is the departure time interval between the front and rear buses.

It can be found from (8) that the bus departure interval is the main factor affecting the number of passengers waiting at the platform. Moreover, the number of passengers on the bus $Q_{bus}^{i}$ can be obtained by analyzing the data on the vehicle’s loading and unloading on the previous platforms. The bus’s maximum passenger capacity $Q_{\max}$ and the predicted number of people departing at the next bus station can be combined to determine the bus remaining capacity $Q_{yu}^{i}$ and the number of passengers that the vehicle can board at the next platform $q_{up}^{i+1}$. Moreover, if $q_{up}^{i+1}\geq Q_{wait}^{i+1}$, it means that all passengers who are waiting $Q_{wait}^{i}$ can embark. Otherwise, it means that the passengers who are waiting cannot all embark, and some passengers will stay on the platform, i.e., $Q_{zhi}^{i}=Q_{wait}^{i+1}-q_{up}^{i+1}$.

Thus, it can be seen that the route parameters of the buses at the *N*-th bus stations are:(9)$$\left\{\begin{array}{c} Q_{zhi}\left(t_{p}\right)=\sum_{i=1}^{N}\left[\max\left(0,Q_{wait}^{i+1}-q_{up}^{i+1}\right)\right]\\ Q_{yu}\left(t_{p}\right)=\sum_{i=1}^{N}\left[Q_{\max}-\left(\sum_{i=1}^{i-1}Q_{up}^{i}-\sum_{i=1}^{i-1}Q_{off}^{i}\right)\right] \end{array}\right.$$

Therefore, from (9), we can see that the bus departure time interval is directly related to the number of people retained on the platform and the free load of the bus line. Moreover, the ideal state is $Q_{zhi}=0,Q_{yu}=0$; therefore, the optimization function for the departure interval can be constructed as:(10)$$\left\{\begin{array}{c} Y_{1}=\min\left[\alpha_{1}\zeta_{1}\left(t_{p}\right)+\alpha_{2}\zeta_{2}\left(t_{p}\right)\right]\\ \zeta_{1}\left(t_{p}\right)=2\left[sigmoid\left(Q_{zhi}\right)-\frac{1}{2}\right]=\frac{1-e^{-Q_{zhi}}}{1+e^{-Q_{zhi}}}\\ \zeta_{2}\left(t_{p}\right)=\frac{Q_{yu}}{NQ_{\max}} \end{array}\right.$$
$\alpha_{1},\alpha_{2}$ are the weight coefficients. The optimal departure time interval can be obtained by the genetic algorithm. Moreover, when $Q_{zhi}=0$, $\zeta_{1}=0$; when $Q_{zhi}=\infty$, $\zeta_{1}=1$; when $Q_{yu}=0$, $\zeta_{2}=0$; when $Q_{zhi}=NQ_{\max}$, $\zeta_{2}=1$.

We use the genetic algorithm to obtain the optimal bus departure time interval, which is introduced in Algorithm 2. It is important to note that in the actual application process, based on the findings of reference [36], which indicates that premature convergence may occur if the prominence of the population continues to increase or remains unchanged, we added population entropy to judge whether there was premature convergence.
**Algorithm 2** Genetic algorithm.**Input I: Qzhi,Qyu,α1,α2****Output I: Optimal target values Y1min,tpbest**
1: α1,α2 need to be defined independently for optimization purposes, and Qzhi,Qyu can be obtained from (9). Moreover, the parameter constraints are Qzhi≥0,0≤Qyu≤Qmax
2: Determine the encoding method and use the real number encoding method.
3: Determine the individual evaluation method; the fitness function is (10).
4: Design a genetic operator, where the selection operation uses a proportional selection operator, the crossover operation uses a single point crossover operator, and the mutation operation uses a basic bit mutation operator.
5: Determine the operating parameters of genetic algorithm M=180, population size G=100, iteration number $P_{e}=0.65$, crossover probability, and mutation probability $P_{m}=0.85$.
end


### 5.2. Station Schedule Optimization

The travel time of the entire bus line is determined by the difference in the arrival time between the first and last bus. According to (1), the travel time $T_{tra}$ of the bus after speed guidance optimization can be obtained:(11)$$T_{tra}=\sum_{j=1}^{i-1}\left(\frac{s_{j}^{1}}{v_{c}^{1j}}+\frac{s_{j}^{2}}{v_{c}^{2j}}+t_{pre}^{j}+\Delta t_{dwell}^{i}+t_{jwait}^{c}\right)$$

From reference [33], we know there is a certain correlation between the bus dwell time and the number of passengers on and off the bus. Therefore, with the decrease in the number of people waiting for the bus, the bus parking time will be optimized to a certain extent, i.e., the value of $t_{pre}^{j}+\Delta t_{dwell}^{i}$. This reduction affects the speed guidance of bus sections overall, even under the condition of punctual arrivals based on the original station schedules. As a result, the traffic efficiency of buses is impacted. Therefore, it becomes necessary to optimize the station schedule of bus lines.

In order to ensure that the overall bus line speed trend is stable, the average line speed was selected to reflect the overall trend; the optimized average line speed should not be lower than the speed before optimization. Therefore, its critical state can be determined to be $\bar{V_{final}}=\bar{V_{usual}}$. $\bar{V_{usual}},\bar{V_{final}}$ are the line average speeds before and after optimization. At this time, according to (12), it is evident that the adjustment amount of the station schedule is influenced by two factors: the optimal amount $\Delta T_{dwell}^{i\rho }$ of dwell time for each shift at each station and the optimal amount $\Delta T_{wait}^{i\rho }$ of waiting time at each intersection; thus, the specific value can be obtained according to the analysis of historical data,
(12)$$\left\{\begin{array}{c} \Delta T_{dwell}^{i\rho }=\bar{t_{dwell(old)}^{i\rho }}-\bar{t_{dwell(new)}^{i\rho }}\\ \Delta T_{wait}^{i\rho }=\bar{t_{iwait(old)}^{c\rho }}-\bar{t_{iwait(new)}^{c\rho }} \end{array}\right.$$

$\bar{t_{dwell(old)}^{i\rho }},\bar{t_{dwell(new)}^{i\rho }}$ are the average station dwell times before and after optimization for the ρ shift of platform *i*, $\bar{t_{iwait(old)}^{c\rho }},\bar{t_{iwait(new)}^{c\rho }}$ are the average intersection times before and after optimization for the ρ shift of platform *i*.

The station schedule is based on a minimum unit of measurement of minutes. Thus, the station schedule time $T_{i\left(new\right)}^{\rho }$ of the ρ shift of vehicles at platform *i* is adjusted to:(13)$$\left\{\begin{array}{c} T_{i(new)}^{\rho }=T_{i(old)}^{\rho }-\Delta T\\ \Delta T=\frac{\Delta T_{dwell}^{i\rho }+\Delta T_{wait}^{i\rho }}{\left|\Delta T_{dwell}^{i\rho }+\Delta T_{wait}^{i\rho }\right|}\left\lfloor \frac{\left|\Delta T_{dwell}^{i\rho }+\Delta T_{wait}^{i\rho }\right|}{30}\right\rfloor \end{array}\right.$$

ΔT is the adjustment amount of the schedule; when the value of $\left|\Delta T_{dwell}^{i\rho }+\Delta T_{wait}^{i\rho }\right|$ is less than 30 s, the schedule will not be adjusted. If it is greater than 30 s and less than 60 s, the schedule will be adjusted by 1 min. If it is greater than 60 s and less than 90 s, the schedule will be adjusted by 2 min, and so on.

## 6. Evaluation

### 6.1. Introduction of Testing Scenario

To test the effectiveness and practicality of the proposed method, an experimental site was selected, namely the bus line in the north and south districts of a science city in Beijing. This bus line has a total length of approximately 4.5 km and consists of 9 bus stations and 8 signal-controlled intersections. The distribution of these stations and intersections is presented in Table 1.

The operation time of the bus line is from 6:30–19:00, which is divided into three periods: morning peak (6:30–9:00), flat peak (9:00–16:30), and evening peak (16:30–19:00); in the morning and evening peaks, the departure time interval is 15 min, in the flat peak, the departure time interval is 30 min. Each platform is equipped with passenger flow detection equipment to monitor the arrival of passengers at the platform in real time. Each intersection signal controller is equipped with a vehicle–road collaborative information interaction device, which can realize the real-time interaction between the intersection signal timing information and the vehicle driving status information.

The vehicles were equipped with card-swiping equipment, speed guidance equipment, and communication devices. These technologies enable real-time interaction with the control center and allow for the collection of passengers’ card data and GPS track data. Moreover, the terminal could push the speed guidance information in real time, which could better carry out the application and implementation of the method in this paper. Moreover, a speed priority coordination-control system of dedicated bus lanes based on passenger demand was developed. The control system structure is shown in Figure 4, which displays the running statuses of bus vehicles in real-time, and provides a strategic assessment and optimization of the speed guidance in a timely manner.

### 6.2. Test Scheme Design

Using the bus departures from the north district to the south district as an example, the departures in the morning peak period include 10 shifts (from the 1st shift to the 10th shift); in the flat peak period, there are 15 shifts (from the 11th shift to the 25th shift); in the evening peak period, there are 11 shifts (from the 26th shift to the 36th shift). The total number of shifts in a day is 36. As illustrated in Figure 5, the bus departure time intervals during the morning and evening peak periods are set at 15 min, while in the off-peak period, the interval is extended to 30 min.

In the optimization process of the bus comprehensive punctuality rate, this paper considers the three indicators of punctual arrival times, the balance of station arrivals, and the mean error of station arrivals to be equally important. Therefore, the weighting factors are set as $\phi_{1}=\phi_{2}=\phi_{3}=1/3$ in (4) is set. In the process of optimizing the bus departure time interval, this paper assigns equal importance to the influence of station passenger flow retention and bus line capacity. Therefore, the weighting factors $\alpha_{1}=\alpha_{2}=0.5$ in (10), combined with the maximum safe load of the bus vehicle (75 people), the maximum passenger flow of $Q_{\max}=75$.

The duration of the test period is 90 working days, with the first 30 working days serving as the baseline test for the state of the bus lines before optimization. The subsequent 30 working days involve testing the implementation of the speed guidance strategy under the existing schedule state. The last 30 working days focus on testing the state of the bus lines after the optimization of the bus line schedule.

### 6.3. Analysis of Test Results

Under stable demand and considering the average speed of the road sections, we conducted an optimization of the bus operation schedule by analyzing the arrival and retention of the platform passenger flow over a period of 60 working days. Figure 5 shows that the station schedule was adjusted for multiple shifts throughout the entire day after the optimization process.

By comparing the bus schedules before and after optimization, it can be observed that the travel time during the morning and evening peak periods has reduced from 28 min to 20 min after optimization. Similarly, the travel time during the flat peak period has decreased from 20 min to 12 min. This indicates that the optimization of the timetable has led to an improvement in the efficiency of bus operations. Additionally, the comparative analysis of schedule optimization and route passenger flow retention is depicted in Figure 6. It can be observed that the optimized departure time interval overcomes the limitations of the original fixed-interval operation schedule, which struggled to accommodate dynamic passenger ride demand. It can increase the departure frequency and reduce the departure interval when the passenger flow demand is high, and appropriately reduce the departure frequency and extend the departure interval when the passenger flow demand is low. Based on an analysis of the bus operation data from 30 working days after the schedule optimization, it was found that the optimization of the bus operation schedule further reduced the passenger flow retention of the bus lines. This led to a 36.22% reduction in the average passenger flow retention of the lines. Additionally, the passenger flow in each shift tended to be stable, further enhancing the service level of bus supply and demand.

In order to quantitatively evaluate the control effect of the dedicated bus coordination-control system at each station and road section in each time period, we collected information on the arrival time, travel time, stop times, and bus carrying capacity at each station for each shift. These data were collected under the demand of bus operation schedule optimization and the control objective of punctual arrival. We conducted a quantitative analysis to compare the effects of the method before and after implementation by comparing bus line travel times, intersection stop times, punctual arrival rates, and other indicators.

There was a significant fluctuation in the average travel time of each shift before the system optimization, as shown in Figure 7. The average error between the travel time of each shift and the set travel time of the same period schedule was as high as 4.40 min. This large error limited the improvement of the operation effect and efficiency. However, after adopting the method of this paper, the average error between the set travel time and the travel time of each shift was 0.08 min, which basically meets the schedule-setting time requirement. After the schedule optimization, the morning and evening peak average travel times were 20.45 min and 11.64 min, respectively, which improved by 27.63% and 61.29%, respectively, before schedule optimization; moreover, the efficiency of the line traffic improved. At the same time, the stop times at the intersections of the line were also optimized, decreasing from the previous 3.99 times per shift to 1.23 times per shift. This represents a reduction of 40.81% and 60.93% compared to before the schedule optimization. This optimization has significantly improved the operation status of the bus line and enhanced the level of service.

The analysis results of the average arrival punctuality rate of each shift are shown in Figure 8. It can be observed that the average arrival punctuality rate of each shift after the speed-guided optimization has reached 83.32%, which can effectively meet the demand for punctual operation control of buses. After the bus schedule optimization, the average arrival punctuality rate of each shift has improved to 90.53%, thereby addressing the low arrival punctuality rate of the original system and reducing the differences between morning and evening peak hours and flat peak hours. Furthermore, after the speed guidance, the average cumulative arrival error of lines in each bus shift decreased from 58.21 s to 20.15 s, indicating a 65.38% improvement in optimization. This enhancement significantly enhances the arrival punctuality service level of buses. With the optimization and adjustment of the bus schedule, the indicator further decreases to 13.25 s, reducing the waiting time of passengers on the platform and ensuring a higher level of punctual service for buses.

## 7. Conclusions

In order to optimize punctuality and the supply–demand service level of dedicated bus lanes, this paper proposes a hierarchical multi-objective optimization model with two layers: speed optimization and schedule optimization. The speed optimization layer involves mathematical modeling of bus operations, a multi-objective speed-decision-making model, and the Lagrange multiplier method for speed optimization. The schedule optimization layer involves optimizing bus departure intervals and station schedules. By optimizing these layers, the goals of operational reliability, comfort, and supply–demand balance were achieved. Practical scenario tests show that the proposed method is effective at improving punctuality and the supply–demand balances. This article presents a fundamental and comprehensive optimization framework to tackle the challenges of weak overall effectiveness in arterial bus priority, considering multiple factors and constraints, and providing a theoretical innovative algorithm for public transportation control. The framework guarantees the punctuality rate and passenger service level of buses and can be further combined with signal priority strategies at intersections, vehicle-to-vehicle communication technology, vehicle–road coordination technology, and collaborative scheduling among vehicles in subsequent research and applications. Its universality ensures adaptability in diverse scenarios and provides reliable support for scholars, managers, and policymakers.

## Figures and Tables

**Figure 1 sensors-23-04552-f001:**
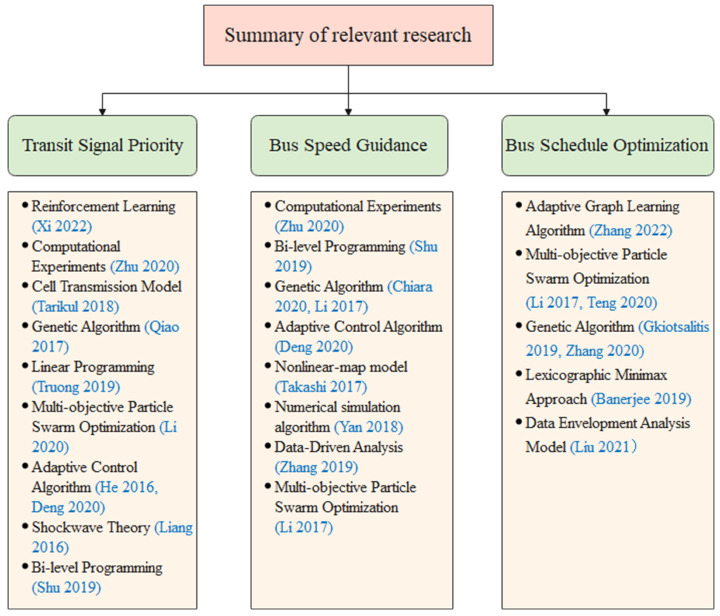
Summary of relevant research [2,3,4,5,6,7,8,9,10,11,12,13,14,15,16,17,18,19,20,21,22].

**Figure 2 sensors-23-04552-f002:**
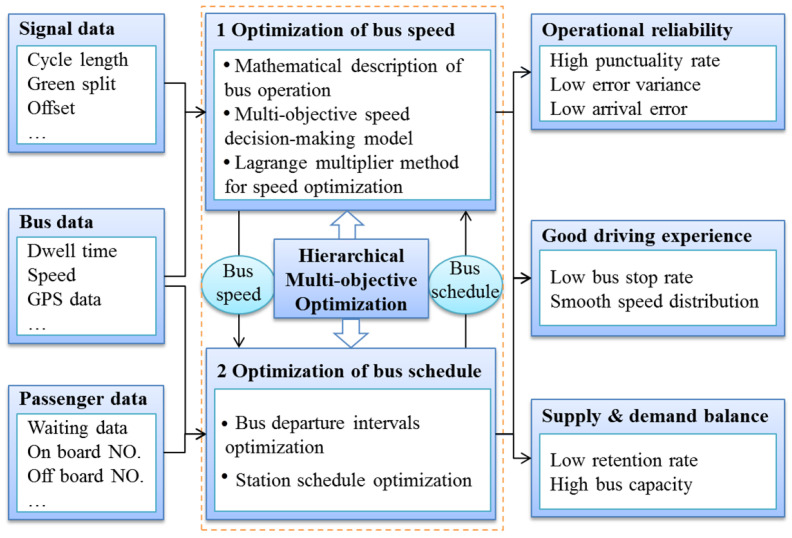
Research framework.

**Figure 3 sensors-23-04552-f003:**
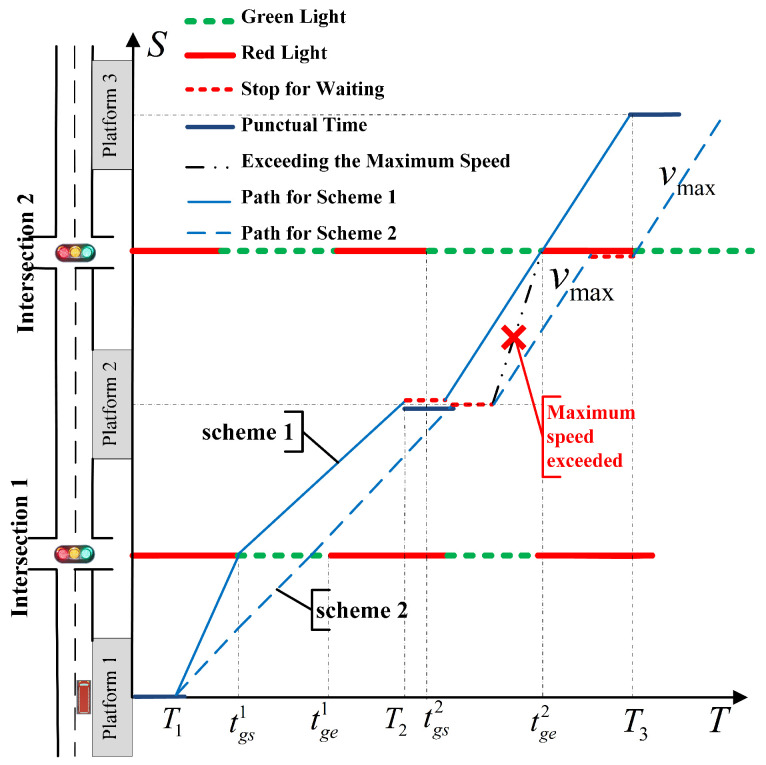
Schematic diagram of bus speed guidance.

**Figure 4 sensors-23-04552-f004:**
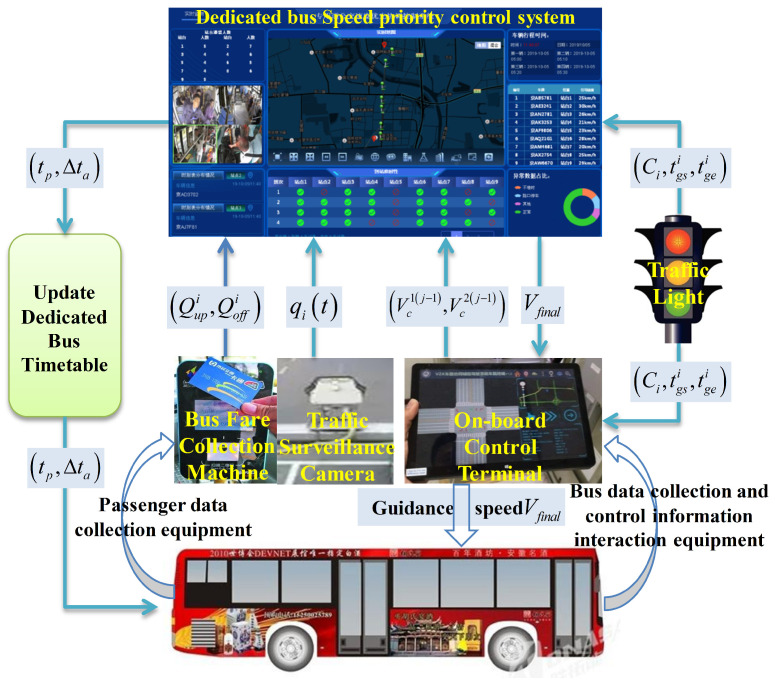
Control system structure.

**Figure 5 sensors-23-04552-f005:**
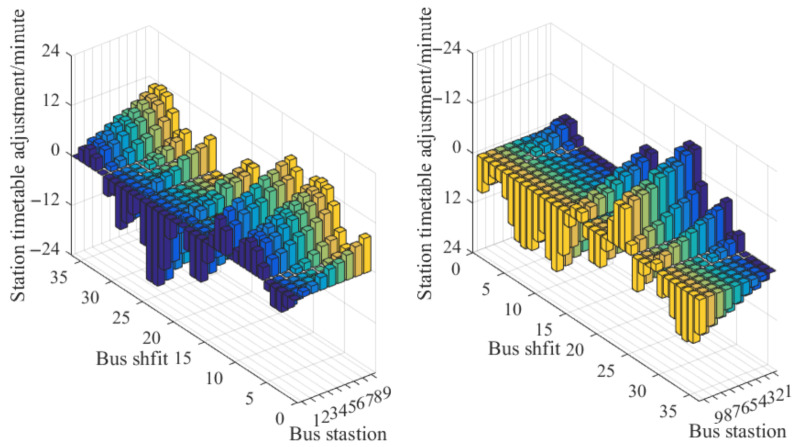
Adjustment after station schedule optimization.

**Figure 6 sensors-23-04552-f006:**
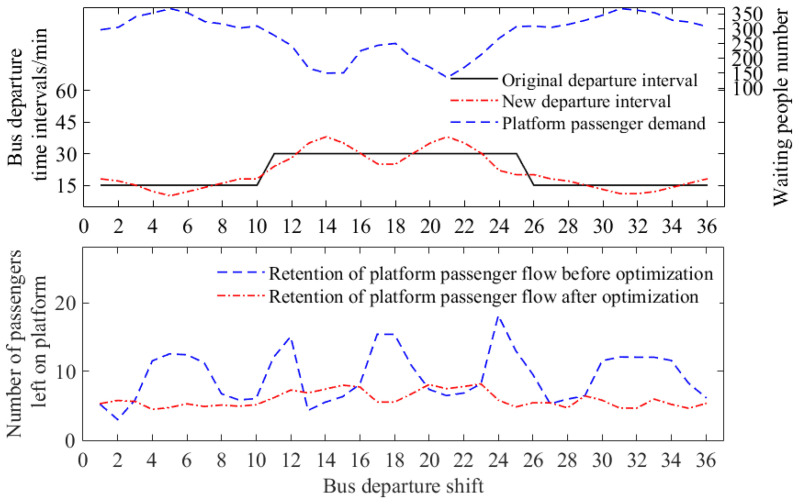
Comparative analysis of schedule optimization and route passenger flow retention.

**Figure 7 sensors-23-04552-f007:**
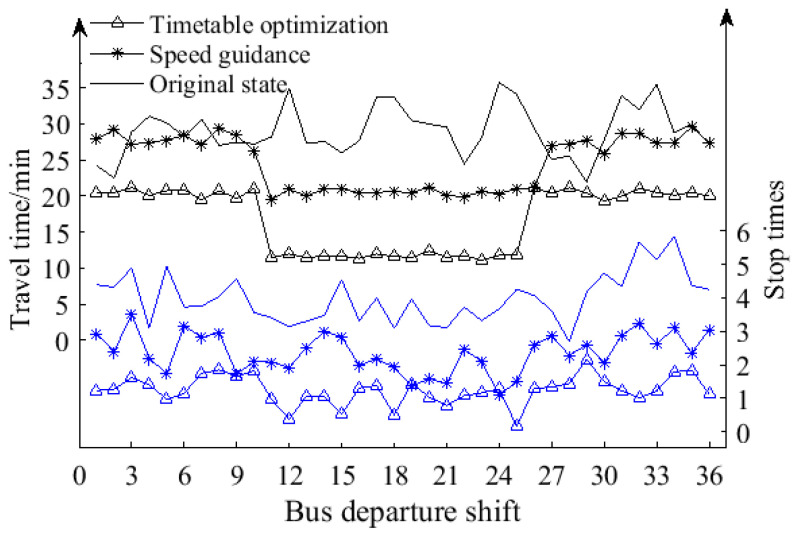
Optimization effect analysis of travel time and stop times at intersections. In the graph, the blue line represents the stop times, and the black line represents travel time.

**Figure 8 sensors-23-04552-f008:**
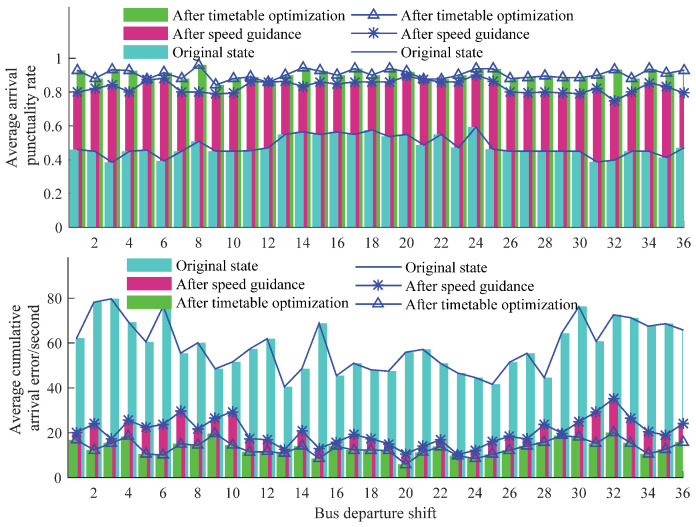
Analysis of the average arrival punctuality rate of each shift.

**Table 1 sensors-23-04552-t001:** Distribution of platforms and intersections.

NO.	Starting Point	Terminal	Distance (m)
1	Platform 1	Intersection 1	260
2	Intersection 1	Platform 2	207
3	Platform 2	Intersection 2	280
4	Intersection 2	Platform 3	195
5	Platform 3	Intersection 3	167
6	Intersection 3	Platform 4	230
7	Platform 4	Intersection 4	170
8	Intersection 4	Platform 5	837
9	Platform 5	Intersection 5	203
10	Intersection 5	Platform 6	236
11	Platform 6	Intersection 6	120
12	Intersection 6	Platform 7	355
13	Platform 7	Intersection 7	295
14	Intersection 7	Platform 8	168
15	Platform 8	Intersection 8	108
16	Intersection 8	Platform 9	678

## Data Availability

The data that support the findings of this study are available from the corresponding author, [Fenghua Zhu], upon reasonable request.

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
