# Peer review of "Hierarchical Multi-Objective Optimization for Dedicated Bus Punctuality and Supply–Demand Balance Control"

_sensors, 2023, doi:10.3390/s23094552_

Round 1
Reviewer 1 Report
While the topic itself is somehow relevant to the journal, the presentation quality is quite high, together with solid technical contribution and interesting case study. However, there are several aspects which can be improved:
Introduction should be extended, as the main part is too short for journal article. Despite that it has all the necessary elements, more elaboration about the challenges addressed, together with practical implication insight should be added
Related works - similar solutions can be summarized within a table somewhere in this section and discussed with respect to issues covered and approaches applied, compared to what was presented in this paper
Optionally, abstract can be revised/extended to be more informative about the approach applied and contributions themselves
Unlike other solutions, it aims to tackle many issues at the same time. The existing researches optimize the bus service level from multiple levels and have achieved certain optimization results. However, the existing punctuality controls are mostly optimized for the single-section level, and fail to optimize the bus punctuality from the perspective of the entire line, which may easily lead to the situation that local optimization but overall poor, thus affect the punctuality service level of buses. In addition, the existing researches seldom pay attentions to the relationship between passenger travel demand and bus capacity allocation, which can easily cause an imbalance between supply and demand and affect the level of bus service
The authors should try to provide some kind of comparison to similar methods, if possible by giving comparative overview of the achieved results for that purpose
Language seems fine, quite clear sentences and easy to follow. No major improvements needed, but spell check suggested just in case that something was missed.
Reviewer 2 Report
The overall idea of the paper (sensors-2337234) entitled “Hierarchical Multi-objective Optimization for Dedicated Bus Punctuality and Supply-demand Balance Control”, by Chunlin Shang et al. is good. The topic that it covers is interesting and being worth of investigation. The main objective of the study is to propose a multi-objective optimization approach for maximising bus service reliability through improving the punctuality and supply-demand service of the dedicated buses. The effectiveness and practicability of the proposed hierarchical multi-objective optimization model are clearly shown. The present work generates new insights on managing Public transportation reliability in busy urban networks.
The document itself is well structured. Furthermore, the purpose of the paper is well stated in the introduction and the findings are clearly and logically presented. Actually good reliable data is included. Accordingly, the scientific value of this submission is high. With these essential features, I think that the paper could be possibly accepted for publication in Sensors Journal (MDPI).
The following are some comments for the authors to consider:
Point 1: Both the abstract and conclusion needs to be carefully revised and improved. The Abstract should be self-contained without referring to the main article.
Point 2: The literature review section is very poorly written and requires a deep improvement.
Point 3: The authors are invited to highlight their contribution to addressing the research gap in the field.
Point 4: Concerning the readability of the manuscript, the authors are invited to keep the sentence simple and straightforward so that it is easy to understand and reflects the crucial findings of the work.
Based on the above comments, I recommend minor revisions.
The overall idea of the paper (sensors-2337234) entitled “Hierarchical Multi-objective Optimization for Dedicated Bus Punctuality and Supply-demand Balance Control”, by Chunlin Shang et al. is good. The topic that it covers is interesting and being worth of investigation. The main objective of the study is to propose a multi-objective optimization approach for maximising bus service reliability through improving the punctuality and supply-demand service of the dedicated buses. The effectiveness and practicability of the proposed hierarchical multi-objective optimization model are clearly shown. The present work generates new insights on managing Public transportation reliability in busy urban networks.
The document itself is well structured. Furthermore, the purpose of the paper is well stated in the introduction and the findings are clearly and logically presented. Actually good reliable data is included. Accordingly, the scientific value of this submission is high. With these essential features, I think that the paper could be possibly accepted for publication in Sensors Journal (MDPI).
The following are some comments for the authors to consider:
Point 1: Both the abstract and conclusion needs to be carefully revised and improved. The Abstract should be self-contained without referring to the main article.
Point 2: The literature review section is very poorly written and requires a deep improvement.
Point 3: The authors are invited to highlight their contribution to addressing the research gap in the field.
Point 4: Concerning the readability of the manuscript, the authors are invited to keep the sentence simple and straightforward so that it is easy to understand and reflects the crucial findings of the work.
Based on the above comments, I recommend minor revisions.
Author Response
Comment 1: “Both the abstract and conclusion needs to be carefully revised and improved. The Abstract should be self-contained without referring to the main article.”
|
Response: Thank you so much for your comments and suggestions. We have made adjustments to the abstract and conclusion according to your suggestion, as follows. Abstract:” Public transportation is a crucial component of urban transportation systems, and improving passenger sharing rates can help alleviate traffic congestion. To enhance the punctuality and supply-demand balance of dedicated buses, we propose a hierarchical multi-objective optimization model to optimize bus guidance speeds and bus operation schedules. First, we present an intelligent decision-making method for bus driving speed based on the mathematical description of bus operation states and the application of the Lagrange multiplier method, which improves the overall punctuality rate of the bus line. Second, we propose an optimization method for bus operation schedules that respond to passenger needs by optimizing departure time intervals and station schedules for supply-demand balance. Finally, experiments are conducted in Future Science City, Beijing, China. The results show that the bus line's punctuality rate has increased to 90.53%, while the retention rate for platform passengers and intersection stop rate have decreased by 36.22% and 60.93%, respectively. These findings verify the effectiveness and practicality of the proposed hierarchical multi-objective optimization model.” Conclusion:” In order to optimize the punctuality and supply-demand service level of dedicated bus lanes, this paper proposes a hierarchical multi-objective optimization model with two layers: speed optimization and schedule optimization. The speed optimization layer involves mathematical modeling of bus operations, a multi-objective speed-decision-making model, and Lagrange multiplier method for speed optimization. The schedule optimization layer involves optimizing bus departure intervals and station schedules. By optimizing these layers, the goals of operational reliability, comfort, and supply-demand balance have been achieved. Practical scenario tests show that the proposed method is effective in improving punctuality and supply-demand balances. This article proposes a fundamental and general optimization framework to address the problem of weak overall effect due to multiple factor constraints in arterial bus priority, providing a theoretical innovative algorithm for public transportation control. The framework guarantees punctuality rate and passenger service level of buses and can be further combined with signal priority strategies at intersections, vehicle-to-vehicle communication technology, vehicle-road coordination technology, and collaborative scheduling among vehicles in subsequent research and applications. Its universality ensures adaptability in diverse scenarios and provides reliable information support for scholars, managers, and policymakers.” |
Comment 2: “The literature review section is very poorly written and requires a deep improvement.”
|
Response: Thank you very much for your suggestions. With your help, we have made revisions to the literature review. However, due to its large size, it is not convenient to attach the revised content after this response. please refer to the revised manuscript where the modifications are highlighted in red. Thank you. |
Comment 3: “The authors are invited to highlight their contribution to addressing the research gap in the field.”
|
Response: Thank you very much for your suggestions. They are very helpful in improving the quality of our article. Following your suggestions, we have made revisions to highlight the research innovation and contributions of our study, as follows: “(1) Existing research on dedicated buses often separates optimization of punctuality service level and supply-demand balance. However, there is a lack of correlation and collaboration between these two goals. To address this issue, a hierarchical multi-objective optimization model has been proposed to optimize the punctuality service level and achieve a balance between supply and demand for dedicated buses. (2) Many existing studies use weighted processing to solve the multi-objective optimization problem, but this approach may lead to the main objective being influenced by other objectives. To address this issue, a multi-objective hierarchical speed decision-making method using the Lagrangian multiplier method has been proposed. This approach first guarantees the comprehensive punctuality rate of the whole line, and then optimizes stop times at intersections and achieves a balanced distribution of guidance speeds. (3) The traditional method of optimizing bus operating schedules based solely on travel time lacks consideration for balancing passenger demand and bus capacity. To address this issue, a bus operation schedule optimization method has been proposed to achieve a balanced relationship between bus capacity and passengers waiting at stations. This approach can overcome inconsistencies between bus schedule settings and passenger travel demand.” |
Comment 4: “Concerning the readability of the manuscript, the authors are invited to keep the sentence simple and straightforward so that it is easy to understand and reflects the crucial findings of the work.”
|
Response: Thank you very much for your suggestions. They have been very helpful in further improving the readability of our article. Following your advice, we have made several revisions to the manuscript. However, due to limitations in the length of this response, please refer to the revised manuscript for the specific changes. We appreciate your suggestions once again. |

Reviewer 3 Report
It seems that the research paper related to 'Hierarchical Multi-objective Optimization for Dedicated Bus Punctuality and Supply-demand Balance Control ' has been conducted systematically.
It would be good if you describe the background of your research and the problems of the target route in the introduction. It would be nice to clarify that the problem is clearly there and that it is part of the solution.
In this study, it is meaningful in that it was applied to the target site through new scheduling and the effect of this was presented. However, it would be nice if you specifically suggested the direction of future research.
In addition, as there is a policy aspect, it would be good to suggest how related research results can be applied to policies.
Review the English text of the entire paper.
Author Response
Comment 1: “It would be good if you describe the background of your research and the problems of the target route in the introduction. It would be nice to clarify that the problem is clearly there and that it is part of the solution.”
|
Response: Thank you very much for your suggestion, which will further improve the content of our article. Based on your advice, we have added a description of the relevant problems in the introduction and further elaborated on how the innovative points of our research can address these issues. The details are as follows: “However, the single punctuality target cannot reflect the reliability and stability of bus, and combined with the literature [2]-[4] we found that multi-objective optimization can overcome the limitations of single-objective optimization, but the multi-objective weighted processing method is easy to reduce the punctuality performance of the line due to taking into account other objectives. And there is also a lack of the studies to consider the whole line punctuality service level and the supply and demand service level.” “(1) Existing research on dedicated buses often separates optimization of punctuality service level and supply-demand balance. However, there is a lack of correlation and collaboration between these two goals. To address this issue, a hierarchical multi-objective optimization model has been proposed to optimize the punctuality service level and achieve a balance between supply and demand for dedicated buses. (2) Many existing studies use weighted processing to solve the multi-objective optimization problem, but this approach may lead to the main objective being influenced by other objectives. To address this issue, a multi-objective hierarchical speed decision-making method using the Lagrangian multiplier method has been proposed. This approach first guarantees the comprehensive punctuality rate of the whole line, and then optimizes stop times at intersections and achieves a balanced distribution of guidance speeds. (3) The traditional method of optimizing bus operating schedules based solely on travel time lacks consideration for balancing passenger demand and bus capacity. To address this issue, a bus operation schedule optimization method has been proposed to achieve a balanced relationship between bus capacity and passengers waiting at stations. This approach can overcome inconsistencies between bus schedule settings and passenger travel demand.” |
Comment 2: “In this study, it is meaningful in that it was applied to the target site through new scheduling and the effect of this was presented. However, it would be nice if you specifically suggested the direction of future research.”
|
Response: Thank you very much for your suggestion. This will help promote the exchange of subsequent research content. Based on your comments, we have added future expectations in the conclusion section as follows: “This article proposes a fundamental and general optimization framework to address the problem of weak overall effect due to multiple factor constraints in arterial bus priority, providing a theoretical innovative algorithm for public transportation control. The framework guarantees punctuality rate and passenger service level of buses and can be further combined with signal priority strategies at intersections, vehicle-to-vehicle communication technology, vehicle-road coordination technology, and collaborative scheduling among vehicles in subsequent research and applications. Its universality ensures adaptability in diverse scenarios and provides reliable information support for scholars, managers, and policymakers.” |
Comment 3: “In addition, as there is a policy aspect, it would be good to suggest how related research results can be applied to policies.”
|
Response: Thank you for your suggestion. Based on it, we have briefly explained the combined process of the public transportation priority policy and our method in the introduction as follows: “With reliable infrastructure investment as a prerequisite, this article's algorithm can optimize bus operations at multiple levels, such as punctuality rate and passenger service level, by monitoring road traffic and bus operation status and combining data edge computing methods. This can increase the proportion of public transportation trips, effectively improve the efficiency and reliability of trunk bus transportation, and thus ensure the reliable promotion and application of public transportation priority policies.” |

Reviewer 4 Report
Thank you for the opportunity to review this interesting paper, which I found as well written and addressing an important topic.
Authors should add a section Discussion.
Discussion and conclusion: Describe in more detail how a new knowledge could be used in practice. I recommend the authors to articulate clearly what the contributions of the paper are to:
1) Theory – the body of conceptual knowledge
2) Practice – to managers / employees / policy makers
Author Response
Comment 1: “Authors should add a section Discussion.
Discussion and conclusion: Describe in more detail how a new knowledge could be used in practice. I recommend the authors to articulate clearly what the contributions of the paper are to:
1) Theory – the body of conceptual knowledge
2) Practice – to managers / employees / policy makers”
|
Response: Thank you very much for your suggestion. It plays a crucial role in the readability and subsequent research of the article. Following your advice, we have added an introduction to the theoretical contribution and application prospects of the article in the conclusion section, as follows: “This article proposes a fundamental and general optimization framework to address the problem of weak overall effect due to multiple factor constraints in arterial bus priority, providing a theoretical innovative algorithm for public transportation control. The framework guarantees punctuality rate and passenger service level of buses and can be further combined with signal priority strategies at intersections, vehicle-to-vehicle communication technology, vehicle-road coordination technology, and collaborative scheduling among vehicles in subsequent research and applications. Its universality ensures adaptability in diverse scenarios and provides reliable information support for scholars, managers, and policymakers.” |
